# Deformation Evolution Law of Surfacing Welding on Thin Bending Plates Based on the Three-Dimensional Thermal Digital Image Correlation Method

Xiqiang Ma [1,2,3], Nan Guo [1,3], Fang Yang [1,2,*], Chunyang Liu [1] and Zhiqiang Guan [4]

1   School of Mechatronics Engineering, Henan University of Science and Technology, Luoyang 471003, China; maxiqiang@haust.edu.cn (X.M.); guonan1860@163.com (N.G.); chunyangliu@haust.edu.cn (C.L.)
2   Longmen Laboratory, Luoyang 471003, China
3   Henan Key Laboratory for Machinery Design and Transmission System, Henan University of Science and Technology, Luoyang 471003, China
4   School of Mechanical and Mining Engineering, University of Queensland, Brisbane, QLD 4072, Australia; guan@uq.edu.au
*   Correspondence: yangfanghkd@haust.edu.cn

**Abstract:** Surfacing on the surface of thin metal bending plates will cause significant deformation, and current numerical simulation and experimental methods cannot fully and truly reflect the deformation state of the bending plate. In this paper, a non-contact detection method based on the three-dimensional (3D) thermal digital image correlation (DIC) method is proposed. The proposed method can be used for the 3D full-field dynamic measurement of metal thin bending plate surfaces. In addition, the evolution law of in-plane and out-of-plane deformation of thin bending plates during surfacing welding and cooling was studied. Moreover, the influence of curvature on the shrinkage deformation of thin bending plate weld was explored, and the correlation between the curvature of thin bending plates and the weld shrinkage was established. Results show that the proposed detection method based on the 3D thermal DIC method can rapidly and accurately detect bending deformation online. The out-of-plane deformation of the surfacing welding of the thin bending plate transits from the disk to the saddle. Furthermore, the curvature of the thin bending plate is inversely proportional to the transverse shrinkage of the weld bead. After the curvature reaches a certain value, it has little effect on the longitudinal shrinkage of the weld bead. This detection method solves the problem of welding deformation simulation verification, truly clarifies the law of welding dynamic deformation, and provides a theoretical basis for welding lightweight manufacturing.

**Keywords:** thin metal bending plates; surfacing welding; deformation evolution law; 3D thermal digital image correlation





## 1. Introduction

The thin bending plate welding structure is widely employed in shipbuilding, automobile, aerospace, and other industries and is one of the main ways to effectively realize lightweight manufacturing. Because welding is a high-temperature forming process, the influence of high temperature, large deformation, and high-brightness arc light must be considered when measuring deformation. As such, current welding deformation detection tests do not yield satisfactory results.

Studies on the deformation of thin bending plates have mainly adopted the finite element method based on the thermo-elastic-plastic theory and the inherent strain theory [1,2]. The calculation of the surfacing welding deformation on metal surfaces is a complex, multidimensional, and multiparameter process. Although the finite element calculation method can deal with various nonlinear problems encountered in the welding process, the comprehensive verification of deformation prediction accuracy is difficult to achieve. Murakawa et al. [3] used the iterative substructure method (ISM) and finite element method to

substantially reduce the number of elements and computation time without loss of accuracy. Ma et al. [4] studied the suppression of the fixture point-solid constraint on the out-of-plane deformation of the weld plate by using the thermo-elastic-plastic finite element method. Kim et al. [5] proposed the inherent strain theory based on the three-dimensional (3D) constraint element and the temperature gradient distribution on the cross-section of the weld to quickly and accurately predict the large welding structure and studied the influence of the point-solid sequence on the deformation of the welded thin plate. Ghafouri et al. [6] adopted the thermo-elastic-plastic finite element method with the double ellipsoidal heat source model and material nonlinear geometric model; they found that the angular deformation and transverse residual stress are greatly affected by external constraints, while the longitudinal stress is less affected. Granell et al. [7] used the thermal-mechanical coupling model based on the birth and death element to accurately predict the welding deformation process. Seo et al. [8] used scalar input variables to modify the traditional inherent strain theory to accurately predict transverse shrinkage and angular deformation and eliminate unnecessary longitudinal forces. Yuan et al. [9] adopted the finite element method based on thermal-structural coupling calculation to study the welding deformation of copper alloy sheets. They found that alternately welding symmetrically from the start and end positions of the weld seam to the middle position of the plate causes the least welding deformation. Shao et al. [10] used artificial neural networks based on FEM to predict multi-objective optimization of MIG Welding and preheat parameters for 6061-T6 Al alloy T-joints and select the optimal combination of process parameters. Waheed et al. [11] studied the effects of thermal and mechanical properties on welding deformation by using the MIG Welding finite element method; the results revealed that differences in the thermal conductivity and thermal expansion coefficient are the main causes of welding deformation. In-depth research has been conducted on improving the finite element model and the convergence method to improve the calculation accuracy and speed of the deformation prediction model of welded thin plates. However, the lack of idealized heat source, molten pool, phase transformation, and other assumptions in the finite element and material performance data result in low-accuracy deformation prediction. Therefore, there are limitations on the prediction accuracy, the acquisition of inherent strain, and the determination of complex welding heat inputs.

In the welding deformation test, high-temperature strain gauges, displacement sensors, surface scanning, and other methods are used for welding deformation detection. Currently, the welding deformation test is only used to verify the accuracy of the finite element prediction of welding deformation [12]. The traditional test method can only measure the deformation state of thin bending plates at a single point and one direction away from the weld bead. Contour surface scanning can only obtain the static full-field deformation before and after welding and cannot track the corresponding deformation points. Therefore, the traditional test method can only ensure the local and static accuracy of the finite element method in predicting the deformation of welded structural parts.

In recent years, the research and application of 3D strain measurement based on the digital image correlation (DIC) method have developed rapidly. The research on the 3D measurement under normal-temperature environments has become increasingly mature and is widely employed in practical engineering fields. Currently, the research on welding deformation using the DIC method is limited to the full-field dynamic deformation measurement outside the high temperature of welding. Strycker et al. [13] used the DIC method and displacement sensors to record the change in the strain during the process of welding a round tube. The results showed that DIC could easily and accurately measure the transient changes in the full-field strain of the specimen. Wang et al. [14] used the DIC method to study welding deformation in small-sized high-temperature steel plates and used a Gaussian lowpass filter to improve the matching accuracy of the DIC method. Yang [15] et al. conducted high-temperature tensile tests on TC4 titanium alloy at 350 °C by using a high-temperature tensile test system with the DIC method. Duan [16] developed a high-temperature speckle production process based on the parametric template and prepared

a high-temperature speckle carrier resistant to temperatures of up to 1200 °C. Chen [17] employed the 3D DIC method for studying the deformation in high-temperature and large-deformation materials and used the high-temperature speckle preparation method and multiple cameras. Jalali [18] used the DIC method to detect the bending creep behavior of materials at 750 °C. Thai [19] studied the effect of image saturation (caused by the luminous intensity of objects) on the DIC measurement accuracy. Ma et al. [20] proposed a correction method for image distortion caused by high-temperature heat flux disturbance. Guo et al. [21] used plasma spraying to produce high-temperature speckles and used the DIC method and the linear polarization filter to test the deformation field of carbon fiber specimens under vacuum at 2600 °C.

In this paper, a non-contact detection method based on 3D thermal DIC is presented to perform 3D full-field dynamic measurement on the surface of thin metal bending plates, reveal the in-plane and out-of-plane deformation evolution law of thin bending plates in the process of surfacing welding and cooling, and explore the influence of the curvature of the thin bending plate on the shrinkage deformation. The findings presented in this paper have important theoretical and application value in the welding industry and are of great significance for lightweight manufacturing, energy saving, and emission reduction.

## 2. Non-Contact Test Method

### 3D Thermal DIC Method

The surfacing welding process involves high-temperature and high-brightness environments. In this paper, the 3D thermal DIC method is proposed to measure the non-contact full-field dynamic deformation of thin bending plates. The thermal DIC method determines the correlation between the reference speckle image and the deformed speckle image to obtain the coordinates after the displacement of the matching point. The basic principle is illustrated in Figure 1. First, speckles are sprayed on the surface of the object before the deformation; the speckle image is then obtained by the camera. To ensure the clarity of the object speckle at high welding temperatures, high-temperature-resistant speckles are sprayed. A rectangular grid centered on point C in the speckle domain is selected as the reference sub-image. During the deformation of the object, the camera continuously tracks the position of the reference sub-image. Finally, the changes in the position of each point in the target sub-image before and after deformation are determined according to the relative position of each point. By using this calculation method, the displacement coordinates of all points in the target image after deformation can be obtained, and the strain and deformation information of all points in the target image after deformation can be calculated using the 3D coordinates of the surface points before deformation.

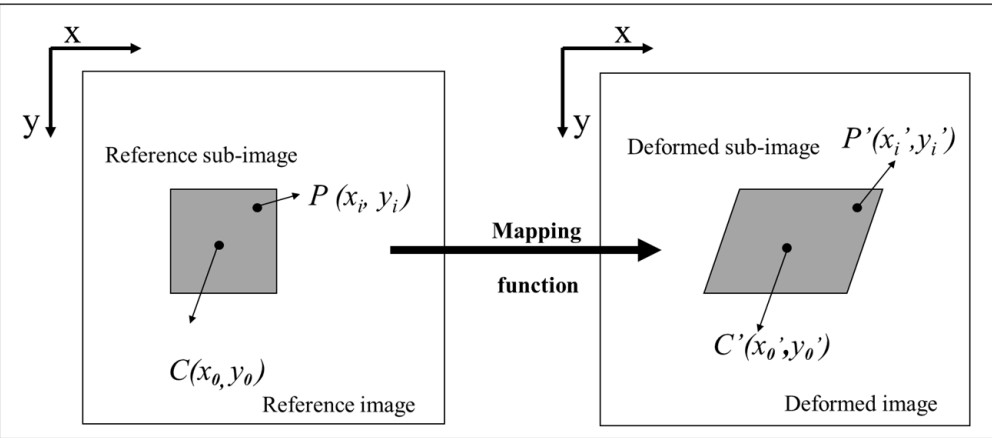

**Figure 1.** Schematic of the thermal digital image correlation method.

It is assumed that the coordinates of any point $P$ in the reference sub-image are $(x, y)$ and the coordinates of the corresponding points $P'$ in the target sub-image after deformation are $P'(x', y')$. If the measured object only has a rigid body displacement, the mapping relationship between the points $P'$ and $P'$ is a zero-order mapping function is:

$$x' = x + u$$
$$y' = y + v \tag{1}$$

In addition to simple rigid body displacement, complex deformation, such as stretching and twisting, may occur when an object is subjected to force. Therefore, when calculating the displacement coordinates of the measuring points on the surface of an object, in addition to considering the changes in the coordinates caused by rigid body displacement, the influence of complex changes such as stretching on the coordinates of the measuring points should be considered. Therefore, the first-order mapping function is introduced:

$$x' = x + u + \frac{\partial u}{\partial x}\Delta x + \frac{\partial u}{\partial y}$$
$$\Delta y y' = y + v + \frac{\partial v}{\partial x}\Delta x + \frac{\partial v}{\partial y}\Delta y \tag{2}$$

where $u$ and $v$ are the displacement of the center point $C$ of the sub-image in the $X$- and $Y$-directions after deformation, respectively, and $\frac{\partial u}{\partial x}$, $\frac{\partial u}{\partial y}$, $\frac{\partial v}{\partial x}$, $\frac{\partial v}{\partial y}$ is the displacement gradient of the reference sub-image.

In the case of more complex deformation, the second-order mapping function can be used to improve the measurement accuracy. In this paper, the first-order mapping function was used for the correlation calculation of displacement coordinates before and after deformation by considering the characteristics and calculation accuracy requirements of the surfacing welding thin bending plate before and after deformation.

### 3. Test Scheme and Discussion of Deformation Evolution Law of Thin Bending Plate Surfacing Welding

*3.1. Surfacing Welding Test Scheme for Thin Bending Plates*

This paper only discusses the evolution law of welding deformation of thin bending plates, and the changes in surface micro-hardness and residual stress of thin bending plates caused by welding are studied in the later stage. Because the bending plate is rolled, the microstructure of the material varies; thus, making the mechanical properties of materials anisotropic; moreover, large residual stress is produced during the manufacturing process. To eliminate the influence of the manufacturing process, in this study, the bending plate was annealed before performing the welding test to eliminate the influence of residual stress on the welding deformation of the bending plate. In the annealing process, the bending plate was heated to 550 °C for 1 h, held for 2–2.5 h, and then cooled to room temperature in the furnace.

The schematic of the thin bending plate welding deformation test device is shown in Figure 2. The convex surface of the thin bending plate was placed on the welding platform. The welding robot, welding torch, and deformation detection device were placed on either side of the thin bending plate. The welding torch was placed above the thin bending plate for surfacing welding, and the detection device was placed below the bending plate (the concave surface of the bending plate was sprayed with high-temperature-resistant speckles). White C3 high-temperature paint was then sprayed evenly on the non-weld area, the leakage plate was covered once dry, and black high-temperature paint was sprayed evenly. Deformation detection was performed. The welding torch was used to perform TIG surfacing welding along the longitudinal center line of the thin bending plate.

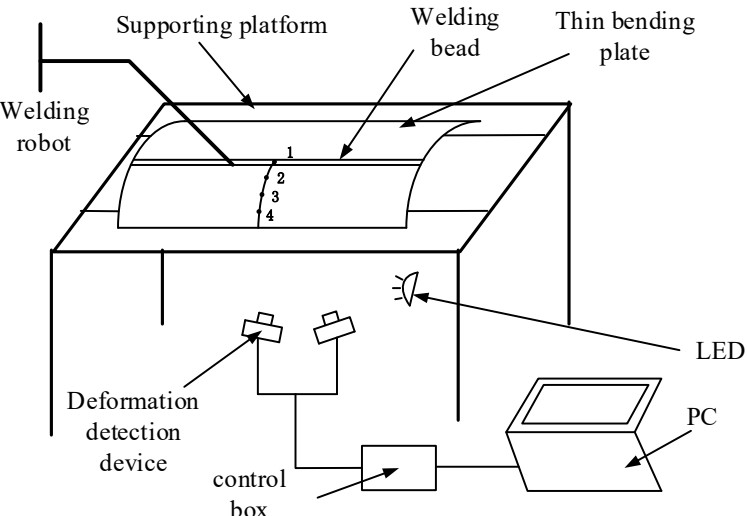

**Figure 2.** Welding and deformation detection in thin bending plates.

C-3 adhesive was used to prepare speckle in the weld and surrounding area. Sandblasting treatment was carried out in the weld area of the curved plate shooting surface to ensure that the area was not reflective, and then the covered leakage plate was affixed, and modulated C-3 glue was evenly smeared to prepare the weld area speckle. In the non-weld zone, white high-temperature paint was uniformly sprayed, the leakage plate was covered after drying, and black high-temperature paint was uniformly sprayed. Finally, the specimen with scattered spots shown in Figure 3 was made. The physical performance parameters of C-3 adhesive are shown in Table 1. Its good heat resistance can ensure that it is not burned and can effectively adhere to the surface of the workpiece.

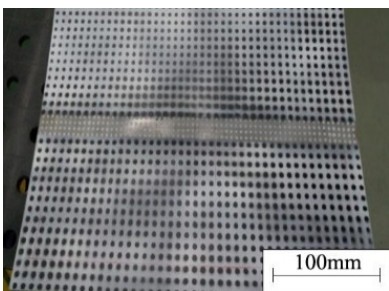

**Figure 3.** Test piece sprayed with high-temperature speckle.

**Table 1.** The physical performance parameters of C-3 adhesive.

| Product | Color | Heat Resistant Temperature (°C) | Electrical Properties | Linear Expansion Coefficient |
|---------|-------|-------------------------------|----------------------|------------------------------|
| C-3 adhesive | gray | 1460 | good heat resistance | Similar to steel ($12.9 \times 10^{-6}$/°C) |

The deformation detection device mainly includes two industrial cameras, a controller, a computer, an LED light source (wavelength 450–460 nm), and a support frame auxiliary device. The position of the camera and the plate in the detection device were adjusted such that the two cameras covered the entire welding plate. The camera acquisition parameters were set, and the light source was adjusted to make the welding speckle image brightness uniform. After camera calibration, the 3D image of the TIG welding and cooling process of the thin bending plate was obtained. The welding time was 60 s, and the cooling time was 360 s. The welding parameters were as follows: thin bending plate material = Q235

(C ≤ 0.18%, Mn 0.35%–0.80%, Si ≤ 0.30%, S ≤ 0.040, P ≤ 0.040), length × arc × width = 300 mm × 200 mm × 3 mm, radius = 500 mm, welding current = 170 I/A, welding voltage = 15 U/V, number of weld layer = 1, welding speed = 5 mm·s$^{-1}$, and argon flow rate = 15 q/L·min$^{-1}$.

The camera taking pictures time and the welding process time were consistent. The camera model was Aca1600–20 gm; the camera resolution was 1626 pixels × 1236 pixels; the pixel size was 4.4 μm × 4.4 μm; the calibration plate field of view was 400 mm × 300 mm; and the camera acquisition frequency was set to 2 pictures per second during the welding process and 1 picture per second during the cooling process. After the welding was completed, the thin bending plate speckle image acquisition was stopped after cooling for 5 min. The dynamic speckle image captured by the camera was imported into the detection software for matching and displacement calculation, as shown in Figure 4. For the full-field grid division of the thin bending plate, the size of the sub-image was set as 30 pixels × 30 pixels, and the step length was 15 pixels × 15 pixels. Stereo matching was performed using the thermal correlation image captured around the same time, and sequence matching was performed using the thermal correlation image captured at different times.

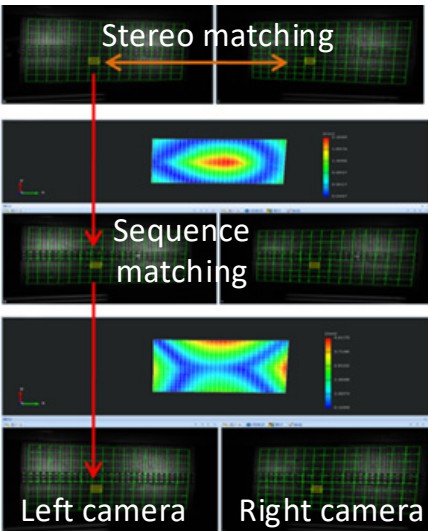

**Figure 4.** Speckle image matching.

### 3.2. Analysis of Out-of-Plane Deformation Evolution of Thin Bending Plate

The out-of-plane deformation cloud atlas of TIG surfacing welding of a thin bending plate with a radius of 500 mm during welding (30 s), at the end of welding (60 s), while cooling (100 s), and when cooling was completed (420 s) is shown in Figure 5. The thin bending plate was convex during the middle of the welding process, the out-of-plane deformation of the whole field was dish-shaped, and the maximum deformation was 2.03 mm. At the end of welding, the thin bending plate was still dish-shaped, but the longitudinal deformation at both ends increased gradually, and the maximum deformation increased to 2.18 mm. After the cooling process was started, the maximum out-of-plane displacement gradually transited to both ends of the bending plate, and the deformation at both ends of the welding bead increased progressively upon moving closer to the width boundary of the bending plate. Furthermore, the middle area of the bending plate welding beam transited negatively to the reverse out-of-plane. With the increase in the cooling time, the deformation at both ends of the welding bead gradually increased, whereas the deformation at both sides of the edge of the thin plate along the length direction was the welding platform. After cooling, the out-of-plane deformation of the bending plate was saddle-shaped, and the maximum deformation increased to 5.40 mm.

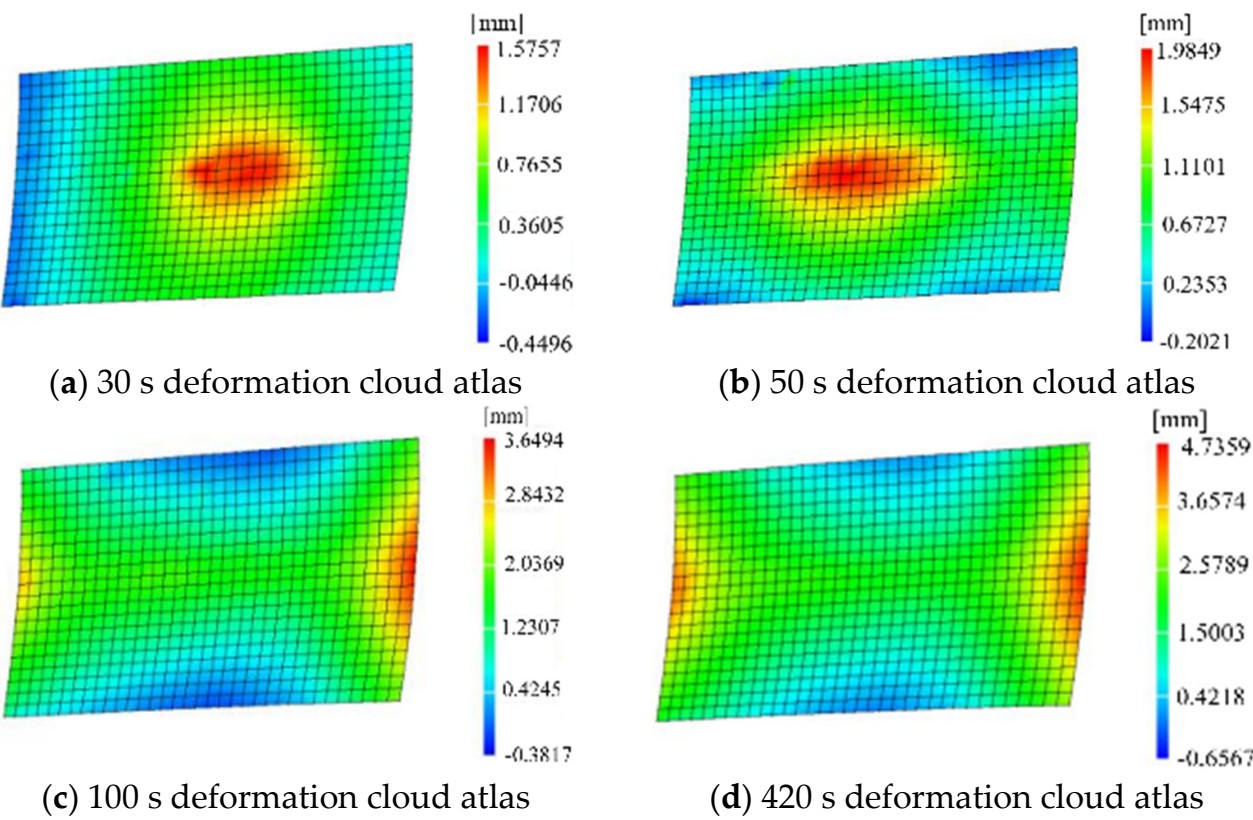

**Figure 5.** Out-of-plane deformation cloud atlas of the thin bending plate. (**a**) 30 s; (**b**) 60 s; (**c**) 100 s; (**d**) 420 s.

The out-of-plane dynamic deformation at the four key points (Figure 2) of the thin bending plate is shown in Figure 6. From the beginning of welding to 50 s, the deformation trend of each point was consistent. All the points underwent maximum deformation relatively quickly. The deformation rate of point 1 was the highest, and other key points decreased in turn. After 50 s, the deformation decreased gradually, but the deformation rate of the points was similar. After cooling (100 s), the deformation of the points was stable. The results showed that during the welding process, the weld bead and the nearby metal thermally expanded and were subjected to the pressure stress of the surrounding metal, making it rapidly convex along the $z$-direction. To coordinate the deformation of point 1, points 2, 3, and 4 also move in the same direction. The closer the point to the weld, the greater the temperature gradient and the more severe the deformation. Upon the completion of the welding (60 s), the thin bending plate began to cool, and the stress of the points gradually changed from compressive stress to tensile stress. However, the welding was not completed at 50 s, and the displacement of the points began to decrease, indicating that point 1 moved farther away from the heat source and the cooling rate accelerated, making the tensile stress exceed the compressive stress and reducing the deformation. After 50 s, a large temperature gradient was observed at the points; thus, the deformation of the test plate was still high at the beginning of cooling. With the decrease in the temperature, the temperature gradient of the weld and its surroundings decreased. Furthermore, carbon steel has a greater elastic modulus at low temperatures; thus, the out-of-plane deformation gradually stabilizes.

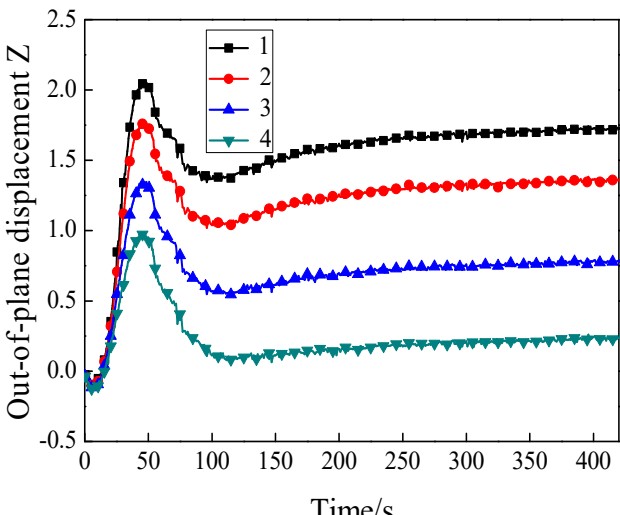

**Figure 6.** Dynamic out-of-plane deformation at key points in the thin bending plate.

Because the welding position was on the convex surface, the compressive stress at the points during the welding process was along the tangential direction; thus, the thin bending plate rapidly became convex, which is related to the initial shape. The out-of-plane deformation evolution of the thin bending plate from dish to saddle was completed during the welding cooling process (100 s).

### 3.3. In-Plane Deformation Evolution Analysis of Thin Bending Plate

The longitudinal and transverse strain distribution on the cross-sectional line of the welding bead after TIG welding and cooling of the thin bending plate is shown in Figure 7. The longitudinal strain on the cross-sectional line of the weld bead was near zero, indicating that the longitudinal plastic deformation on the cross-sectional line of the bending plate weld was small. The transverse plastic strain was negative in the weld area, indicating that the shrinkage deformation occurred after the weld cooling, and the compressive plastic strain was the highest at the center of the weld. In the area adjacent to the weld, the negative plastic strain gradually decreased and transited to positive strain. Therefore, the welding plastic region of the thin plate includes the negative compressive plastic strain region and the positive tensile plastic strain region.

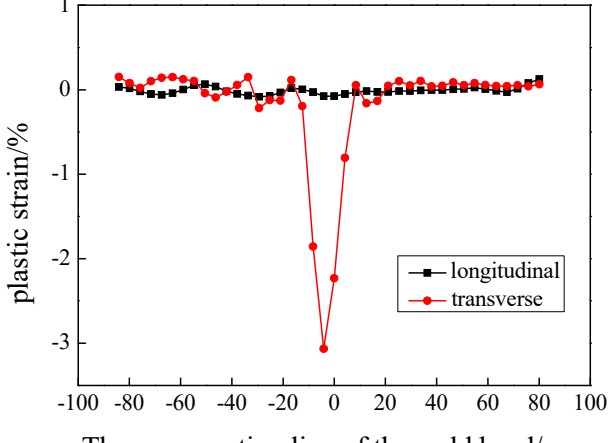

**Figure 7.** Plastic strain distribution on the cross-sectional line of the thin bending plate weld.

In the weld zone, due to the solidification of the liquid metal after cooling, thin plate transverse shrinkage occurred, and the maximum shrinkage was observed at the center of the weld. In the heat-affected zone, due to the tensile stress caused by the cooling of the weld and the influence of its own thermal expansion, tensile plastic deformation occurred in the transverse direction, and the transverse strain was positive. Due to the low temperature gradient and the small tensile stress in the area outside the heat-affected zone, the plastic deformation in the transverse direction was very small, and the transverse strain fluctuated near zero. The longitudinal strain on the transverse line of the weld fluctuated due to the flow of the molten pool around the weld and was close to zero. Therefore, the plastic deformation along the transverse line of the weld is mainly transverse compression deformation which is mainly caused by the uneven shrinkage of the weld and the surrounding metal during cooling.

During the welding process, the weld bead and the metal in the heat-affected zone undergo thermal expansion. When the thermal stress exceeds the yield stress of the material, tensile plastic deformation occurs. When the cooling process is initiated, the transverse shrinkage gradually overcomes the tensile strain and finally produces the negative shrinkage strain. In the area where the tensile plastic deformation has occurred relatively far from the weld, the tensile plastic deformation exists near the compression plastic deformation area due to the failure of the cooling shrinkage to completely balance the tensile strain.

### 3.4. Shrinkage Analysis of Thin Bending Plate Welds with Different Curvatures

By using the same welding deformation test parameters, the influence of different curvatures of thin bending plates on weld shrinkage was studied. The radius of the thin bending plate was selected as 100, 200, 300, 500, 1000 mm, and flat plate. The influence of the boundary effect was ignored in thin bending plate weld bead; the longitudinal and transverse shrinkage of weld with uniform shrinkage on the center line of weld can be expressed as follows:

$$S_L = \sum_{i=1}^{n-1} (x_{i+1} - x_i) \tag{3}$$

$$S_T = \sum_{i=1}^{n-1} (y_{i+1} - y_i)$$

where $S_L$ is the longitudinal shrinkage of weld, $S_T$ is the transverse shrinkage of the weld, $n$ is the number of grid points uniformly distributed in the center of the weld, $x_i$ is the longitudinal coordinates of the weld centerline point $i$, and $y_i$ is the transverse coordinates of the weld centerline point $i$.

The relationship between the longitudinal shrinkage and the curvature of the thin bending plate is illustrated in Figure 8; the longitudinal shrinkage of the thin bending plate with curvature was found to be smaller than that of the flat plate, and the weld shrinkage was only 0.15–0.2 mm indicating that under the same welding and design parameters, the initial transverse curvature has little effect on the longitudinal shrinkage under longitudinal convex welding.

Because the longitudinal shrinkage of the weld is caused by the longitudinal shrinkage force along the weld direction, the longitudinal shrinkage force depends on the welding process and material properties and has little correlation with the geometric size of the welded joints and components. Therefore, under the same welding conditions, the longitudinal shrinkage forces of bending plates with different radii are approximately equal; thus, the difference in the longitudinal shrinkage is relatively small.

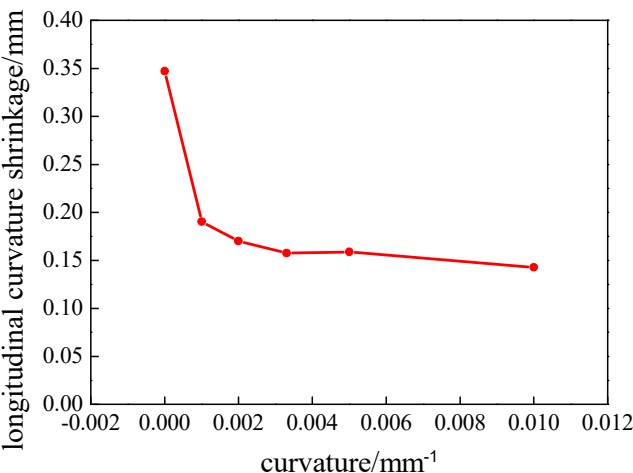

**Figure 8.** Relationship between curvature and longitudinal shrinkage of welds.

The transverse shrinkage in thin bending plate welds is caused by the transverse shrinkage force, which is not only related to the welding process and material properties but also to the cross-section size, material stiffness, and constraint of the test plate. The larger the curvature of the bending plate, the greater the material stiffness and the constraint between the matrix, and the smaller the transverse shrinkage force and the transverse shrinkage. The welding position has little effect on the transverse shrinkage of the bending plate, whereas the effect of curvature is relatively large. The relationship between the transverse shrinkage of the weld and the curvature of the thin bending plate is shown in Figure 9. The fitting of the transverse curvature and the transverse shrinkage of the weld is linear, as can be noted from the following expression:

$$S_T = -20.259 C_{curvature} + 0.418 \qquad (4)$$

where $C_{curvature}$ is the transverse curvature of the weld.

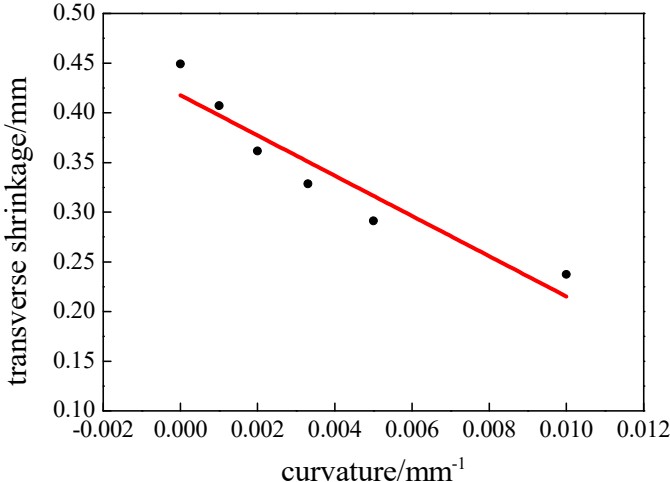

**Figure 9.** Relationship between curvature and lateral shrinkage of welds.

With the increase in the transverse curvature of the thin bending plate (decrease in the radius), the transverse shrinkage of the weld decreases linearly in inverse proportion indicating that the initial transverse bending shape of the thin bending plate greatly affects the transverse shrinkage of the weld after welding deformation. The greater the curvature, the greater the transverse stiffness of the bending plate and the smaller the deformation.

## 4. Conclusions

In this paper, the in-plane and out-of-plane deformation evolution of thin bending plates during the TIG welding and cooling process was studied using the 3D thermal DIC method. The following conclusions were obtained:

(1) The out-of-plane deformation of thin bending plate surfacing welding is characterized by welding deformation evolution from disk-shaped during the welding process to saddle-shaped after cooling.

(2) The maximum transverse shrinkage deformation occurs on the cross-sectional line of the weld bead. The larger the bending radius, the greater the transverse shrinkage deformation. Curvature has little effect on longitudinal shrinkage but has a greater effect on transverse shrinkage, and the welding position has little effect on transverse shrinkage.

(3) It provides an accurate and efficient solution for the study of sheet instability and deformation under high temperature and strong light conditions and provides a theoretical basis for revealing the mechanism of surfacing deformation of thin bending plates.

**Author Contributions:** Conceptualization, X.M.; methodology, X.M. and N.G.; software, F.Y.; validation, X.M. and N.G.; investigation, C.L.; resources, X.M.; data curation, N.G.; writing—original draft preparation, F.Y.; writing—review and editing, Z.G.; supervision, C.L.; project administration, X.M.; funding acquisition, F.Y. All authors have read and agreed to the published version of the manuscript.

**Funding:** This research was funded by the National Key R&D Program of China (Grant No. 2020YFB2009602); the Henan Province Key Science and Technology Project (Grant No. 202102210263); and the Key Science and Research Program of the University of Henan Province (Grant No. 21A460014).

**Institutional Review Board Statement:** Not applicable.

**Informed Consent Statement:** Not applicable.

**Data Availability Statement:** Data is contained within the article.

**Conflicts of Interest:** The authors declare no conflict of interest.

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
