# Peer review of "Deformation Evolution Law of Surfacing Welding on Thin Bending Plates Based on the Three-Dimensional Thermal Digital Image Correlation Method"

_coatings, doi:10.3390/coatings12081084_

Round 1

Reviewer 1 Report

Overall a good paper.

However, the article should make a clear distinction between Intro, Methodology and Results and Discussion. I suggest 2. should be methodology and 3. should be results and discussion. 

Other minor comments can be seen in the attached file.

Author Response

The article has been revised according to the suggestion and has been clear distinction between Intro, Methodology and Results and Discussion.

Reviewer 2 Report

Comment No. 1: The Abstract should contain answers to the following questions: What problem was studied and why is it important? What methods were used? What are the important results? What conclusions can be drawn from the results? What is the novelty of the work and where does it go beyond previous efforts in the literature? Please include specific and quantitative results in your Abstract, while ensuring that it is suitable for a broad audience. References, figures, tables, equations and abbreviations should be avoided.

Comment No. 2: The originality of the paper needs to be stated clearly. It is of importance to have sufficient results to justify the novelty of a high-quality journal paper. The Introduction should make a compelling case for why the study is useful along with a clear statement of its novelty or originality by providing relevant information and providing answers to basic questions such as: What is already known in the open literature? What is missing (i.e., research gaps)? What needs to be done, why and how? Clear statements of the novelty of the work should also appear briefly in the Abstract and Conclusions sections.

Comment No. 3: An updated and complete literature review should be conducted and should appear as part of the Introduction, while bearing in mind the work's relevance to this Journal and taking into account the scope and readership of the journal. The results and findings should be compared to and discussed in the context of earlier work in the literature.

Comment No. 4: This paper should be edited grammatically.
Comment No. 5: Result and discussion section can be more improved from the physical point of view.

Comment No. 6: What is your main novelty of your problem?

Reviewer 3 Report

The authors have done a wonderful job of explaining the in-plane and out-plane deformation behaviour through DIC images. However, a Few of my suggestions to improve the manuscript up to the reader's interest

1. While describing the in plane and out plane deformation behaviours, authors need to investigate the surface integrity properties as well. Like Microhardness variation through the thin plane.

2. There must be residual stress present on the welded surface.. needs to explain about its nature either it is tensile in nature or compressive..

Both the above comments are required as authors are talking about deformation. Then it must cover the several aspects of deformation.

3. Results of in-plane and out-plane deformation must be supported by some recent citations in a comparative way.. there are various studies already available on it.

4. At last conclusion should include some future prospects.

Reviewer 4 Report

Manuscript ID: coatings-1793728 entitled:

Deformation Evolution Law of Surfacing Welding on Thin Bending Plate based on Three-dimensional Thermal Digital Image Correlation Method

Authors:  Xiqiang Ma , Nan Guo , Fang Yang * , Chunyang Liu , Zhiqiang Guan

General comment

The paper presents a study regarding a non-contact detection method based on the three-dimensional (3D) thermal digital image correlation (DIC).

Some recommendations and observation remain:

1. Speckle Pattern Preparation is recommended to be insert (surface preparation method) as is necessary to ensure a stable adherence of the coating to the specimen. Specify other parameters: the nature of the plates (Q235 is a plain carbon structural steel, max 0.17% C), electrodes used in welding process, number of weld layer, wire feed rate etc.

2. In this study   filters or illumination (illumination and filters to minimize the emitted light) were used?

3. The legend of Figures 4 is difficult to read and it would be better to increase the font or accuracy of the images.

4. In Figure 6, insert the legend in English. Check upper or lower subscript of the units (where is necessary) (at R183 welding speed = 5 v/ mms-1). For clarity is better to write: welding speed (v) = 5 mm.s-1.

5. The welding heat input is affected by the change of arc length ( literature mentioned that a decrease of arc length by ~2 mm is able to decrease the welding power with~ 10%, for a constant current power source). The instability of welding arc may lead to a poor welding quality, can authors comment about the quality of welding?

6. The terms in Eq 4 must be describe (ST represent …. and R -…..)

7. The authors write that the fitting of the transverse curvature and the transverse contraction of the weld is linear, as can be seen from Equation 4. Check Equation 4 (for the R data shown in Figure 8, different ST values ​​are obtained using this. Eq). Enter the goodness of the fit for linear regression.

Round 2

Reviewer 2 Report

Now it can be published.

Author Response

Thank you for your valuable comments

Reviewer 4 Report

Accept in present form.

Author Response

Thank you for your valuable comments